# Geriatric Oncology in Portugal: Where We Are and What Comes Next—A Survey of Healthcare Professionals

**DOI:** 10.3390/geriatrics7050091

**Published:** 2022-09-06

**Authors:** Joana Marinho, Sandra Custódio

**Affiliations:** 1Medical Oncology Department, Centro Hospitalar Vila Nova de Gaia/Espinho, 4434-502 Vila Nova de Gaia, Portugal; 2Associação de Investigação de Cuidados de Suporte em Oncologia (AICSO), 4410-406 Vila Nova de Gaia, Portugal

**Keywords:** geriatric oncology, older adults, geriatric assessment, healthcare policies

## Abstract

In keeping with the trend worldwide, in Portugal, more than 60% of newly diagnosed patients with cancer are aged 65 years or older, which makes older adults the most common population seen in an oncology practice. This study’s objectives were to assess geriatric oncology practices in Portugal and investigate medical professionals’ current needs and perceptions on the treatment of elderly cancer patients. Methods: A cross-sectional study was conducted using a web-based survey of healthcare providers treating elderly patients. Results: There were 222 responses: 62.6% of physicians reported the absence of geriatric oncology and/or geriatrics consultations in their institutions, 14.9% had guidelines for the management of older patients with cancer and 4.5% had physicians dedicated to geriatric oncology. The reported use of geriatric assessment tools was 23.4%. Medical oncologists and physicians from medical specialties (*p* = 0.009) and those practicing in the south of Portugal (*p* = 0.054) were more likely to use geriatric assessment. Education and training in geriatric oncology was identified by 95.0% of respondents as an unmet need. The inquiries identified that geriatric assessment could be useful to define a therapeutic strategy (85.1%), detect frailty (77.5%), predict toxicity and improve quality of life (73.4%). Conclusions: There is a paucity of expertise and training in geriatric oncology in Portugal but an increasing perception of the value of geriatric assessment and the demand for education. In the next years, Portugal will progress in this area with the aid of the recently created Geriatric Oncology Working Group.

## 1. Introduction

Aging is one of the strongest and most predictable risk factors for the development of cancer. The pool of patients over the age of 65 years being diagnosed with and surviving their cancer is rapidly expanding [1]. According to GLOBOCAN 2020, in Portugal, more than 60% of patients who are newly diagnosed with cancer are aged 65 years or older, which makes this the most common population seen in an oncology practice [1]. Older adults with cancer are heterogeneous and have wide variability in their health status and social support; therefore, they require a personalized approach to cancer therapy. As healthcare systems remain single-disease focused, the optimal healthcare pathway for multimorbid patients is very complex [2]. Optimal delivery of care in this population faces multiple hurdles related to high cost, lack of logistical resources and lack of evidence-based care [2,3,4,5]. These challenges often lead to over-treatment, under-treatment or suboptimal outcomes. Aging is an individual process with increasing variation in comorbid disease. Therefore, chronological age, used for older patient stratification in oncology, often poorly correlates with biological age and functional status in this population. Geriatric assessment (GA) of older adults with cancer before the initiation of anticancer treatments is recommended by international guidelines [6,7].

GA is a multidimensional, interdisciplinary diagnostic process with a focus on medical, physiological, functional and psycho-social domains, employed in older vulnerable or frail patient populations, in order to identify impairments that are not routinely detected during usual oncology consultations. It also includes a coordinated and integrated plan for treatment and follow-up [8,9]. Identifying impairments through GA allows the implementation of personalized interventions resulting in several substantial benefits and improved outcomes [9]. A multidisciplinary consultative GA, a geriatrician embedded within an oncology clinic and primary management by a dual-trained geriatric oncologist are just a few examples of the various models for integrating geriatrics into oncology care that exist (reviewed in [5]). The ideal model does not exist, and GA should be flexible and tailored to the resources available. Multiple randomized controlled trials unequivocally demonstrated the benefits of GA and GA-guided interventions in reducing the toxicity of systemic treatments and improving patients’ health-related quality of life [10,11,12,13,14]. Whether and how often cancer providers use GA tools when treating older adults is not known, but a strong association was found between awareness of the ASCO Guidelines and the use of GA in practice [4,7]. GA is considered complex and resource demanding. Thus, the implementation is a challenge, especially in areas and practices with limited time, training and resources. In addition, relatively few geriatric specialized care providers exist to facilitate such assessments [15]. In Europe, there are countries where geriatrics is a specialty or a sub-specialty and others where it is a competence. In Portugal, geriatrics is a competence, and few geriatricians exist [16].

The aim of this work was to examine the position of geriatric oncology practice in Portugal and analyze medical professional’s current needs and perceptions in the management of older adults with cancer in our country.

## 2. Materials and Methods

### 2.1. Survey Development and Setting

A 10-question, online, web-based survey was prepared using the free server Google Forms (https://docs.google.com/forms, accessed on 13 August 2022) (Table 1 and Appendix A). The questionnaire was anonymous and did not collect personal data other than age and gender. From September to October 2019, the survey was shared by the Portuguese Oncology Society mailing list, which included 700 members from different medical specialties at that time. First, a brief introduction outlining the main goals of the survey was presented. In order to submit the survey and be included, responses to all 10 questions were required. Respondents who provided consent to participate and responded to the full questionnaire were included.

Questionnaire domains included the following respondent characteristics (gender, age, work geographic location, medical specialty). No financial incentives were offered to respondents. The time to complete the survey was on average 5 min. 

### 2.2. Data Analysis

Descriptive analyses were conducted for responses to survey questions. When the questionnaires were sent in by the respondents, data were automatically saved in an Excel sheet (each question in separate columns) providing a database for analysis. Data were analyzed using IBM^®^ SPSS v24.0. Descriptive statistics were presented as frequencies (n) and percentages (%) for categorical variables and as medians and range for continuous variables. A chi-square (χ^2^) test of independence was performed to examine the relationship between geographic differences and answering to “No” to questions 1 to 3. Age, location and gender differences were also explored for questions 5–8 using the χ^2^ test. *p* < 0.05 was considered as statistically significant.

## 3. Results

### 3.1. Facts and Figures—Policies for Older Adults with Cancer

A total of 222 physicians from different medical specialties completed the survey; the majority of respondents were medical oncologists (n = 119, 53.6%) (Table 2). There was a good geographical representation of the country (Table 2). When questioned about the existence of specific consultations or specialists for the care of elderly cancer patients, as well as management protocols for this population, 62.6% of physicians stated their institutions did not offer consultations in geriatric oncology and/or geriatrics (Figure 1A), and among the 12.6% of cases that did, geriatrics made up the majority (12.1%) (Figure 1A). Only 4.5% of respondents reported the existence of a physician dedicated to geriatric oncology in their hospitals (Figure 1B), and 14.9% of institutions provided specific guidelines for the management of older patients with cancer (Figure 1C). The majority of respondents (92.8%) perceived an increase, in clinical practice, in number of elderly cancer patients, and almost all (98.2%) admitted that older adults with cancer require different care than younger ones (Appendix A). A significant association between the absence of geriatric oncology practices and location of the practice within the country was found, with 77.1% of physicians from the north of the country answering “No” to the first three questions vs. 62.0% in the south, where the Portuguese capital is located (*p* = 0.033).

### 3.2. Geriatric Assessment and Screening

To support treatment decisions and evaluate older adults, 82.4% of clinicians sense a need for additional scales other than ECOG-Performance Status (ECOG-PS) and Karnofsky Performance Scale (KPS), and 12.6% have never thought about this subject (Appendix A). When asked about the use, in clinical practice, of geriatric assessment/screening tools to evaluate elderly patients, only 23.4% (n = 52) reported using these tools (Figure 2A). Medical oncologists and physicians from medical specialties (*p* = 0.009) and those practicing in the south of Portugal (*p* = 0.054) were more likely to report performing a GA (Table 3). The most listed tools reported by the respondents were the geriatric screening tools, Geriatric 8 (G8) questionnaire [17] and Vulnerable Elders Survey-13 (VES-13) [18,19] (n = 18 out of 47 responses), and tools to assess activities of daily living and instrumental activities of daily living (with Barthel and Katz indexes, respectively) (n = 13 out of 47) (Figure 2B). Almost all the professionals (95.0%) considered that there is a need for more education regarding geriatric oncology as well as additional training in this field, and 5.0% had never thought about this subject. In terms of age or gender, we found a high homogeneity among answers to questions 5 to 8, with no significant differences.

### 3.3. Decision-Making and Active Aging Initiatives and Policies

The majority of the respondents (85.1%) considered that GA could be useful in guiding/defining a therapeutic strategy: 77.5% to detect frailty and 73.4% to predict toxicity and improve quality of life (Table 4). Regarding what is important to develop in the field of geriatric oncology in the country, 80.2% considered that GA should be done systematically at oncology departments. Education and training on the needs of elderly cancer patients, both at the undergraduate level and in advanced training, was identified as essential by 70.3%. Other suggestions were regarding geriatricians being a part of multidisciplinary teams (47.7%) and the creation of study groups and/or geriatric oncology units (45.9% and 30.6%, respectively).

## 4. Discussion

In this study, we set out to explore the current geriatric oncology practices in Portugal and analyze medical professionals’ current needs and perceptions in the management of older cancer patients. To our knowledge, this is the first survey of its kind in our country, as there were no data reported on this subject so far. Other countries, where knowledge and practices in geriatric oncology were examined by nationwide surveys, revealed a high demand for education, with an overall acceptance that the GA is an evidence-based way to evaluate older adults with cancer, though it is still less frequently used than anticipated [4,20,21,22].

Interestingly, 62.6% of respondents stated that there was no geriatric oncology care in their hospitals, and only a few (12.6%) reported having care in geriatrics. There was also a limited number of institutions that provided specific guidelines for the management of older patients with cancer. One of the possible explanations is the limited number of geriatricians available in the country (n = 64) [16]. In Portugal, geriatrics is not recognized as a medical specialty but as a competence by the Portuguese Medical Association, and this recognition is relatively recent (since 2014) [16]. As a consequence, the number of older adults per geriatrician in Portugal is considerable and estimated to be 31.590 [23]. In Spain, according to the Spanish Society of Geriatrics and Gerontology, there are approximately 2456 physicians involved in geriatric care (970 are geriatricians), and 33 accredited centers to train geriatricians [23,24].

According to Statistics Portugal, there was a 4.4% increase in the last 10 years in the population aged 65 years or older in Portugal, and considering the 2.1% decrease in the total population, the absolute number of older adults is larger, making Portugal one of the most aged countries in Europe [23,25]. This is in line with respondents’ perceptions, as the majority (92.8%) reported an increase in the number of elderly cancer patients in clinical practice.

Almost all the respondents (98.2%) admitted that older adults with cancer require different care than younger patients, and 82.4% of clinicians recognized that ECOG-PS and KPS, used in clinical practice to support treatment decisions, are not enough. These findings are significant because there is increasing evidence that conducting a GA can lead to adaptations in clinical management and improved outcomes for older adults with cancer [26]. However, only a small percentage of respondents (23.4%) reported using GA and/or geriatric screening tools to assess elderly patients in clinical practice, and surprisingly, 8.1% had never heard of GA. The proportion of physicians who reported performing a GA is comparable to that found in other studies, such as in a survey of cancer providers in the United States (U.S.) (21%) [4] or a nationwide survey in Mexico (18.9%) [20]. Studies from European countries, such as Spain, revealed a 31% use of GA in clinical practice [21], and an older study from the Netherlands showed that 60% of healthcare professionals performed some sort of geriatric evaluation [27]. Because Europe has the largest population share of older adults worldwide, the increasing awareness of the need for geriatric oncology practices in this continent is not surprising.

In the south of Portugal, where Lisbon the capital is located, there was a significantly higher percentage of physicians reporting the use of GA, which might be related to the relative higher number of geriatric and geriatric oncology practices and/or protocols implemented, when compared to other locations in the country. This reflects that there is an urgent need to increase awareness of the growing needs of older patients with cancer and guidelines available, and to implement healthcare policies aimed to improve their care. Global geriatric oncology initiatives are revolutionizing the way elderly cancer patients are being treated [2,28], particularly in Europe, where there have been a rising number of new specialized clinics, initiatives to enroll in clinical trials and joint initiatives to develop clinical trials for older adults with cancer [3].

Despite its uncommon usage, the majority of respondents considered that GA would be helpful for guiding treatment decisions, identifying frailty, predicting toxicity and improving quality of life. Thus, GA must serve as the focal point of the decision-making process [29].

The most crucial issues identified, in terms of what needs to advance and improve in the field of geriatric oncology in the country, were GA systematically performed at oncology services (80.2%) and improvement in education and training on the needs of elderly patients (70.3%). Other suggestions included geriatricians being a part of multidisciplinary teams and the creation of study groups and/or geriatric oncology units. Most of the needs in geriatric oncology identified by Portuguese respondents are common among European countries, as recently revealed by an ESMO–SIOG Joint Working Group short survey on the management of older patients with cancer [30].

In Portugal, there are four centers offering postgraduate geriatrics training, although there are no specific programs toward training in geriatric oncology. There is also a lack of training at the undergraduate level. Therefore, there is a need to disseminate knowledge and integrate geriatric oncology in the curricula for healthcare professionals’ education, in order to develop a workforce in the field and in the context of a broader implementation of active aging-related policies and initiatives.

As a consequence of this growing need, in 2020, the Portuguese Oncology Society established the Geriatric Oncology Working Group (GTOG), with the aim to improve knowledge and develop cooperative clinical, educational and research initiatives in geriatric oncology [31]. Several partnerships have been established so far, with the International Society of Geriatric Oncology (SIOG), Oncogeriatrics Group of the Spanish Society of Medical Oncology (SEOM) and with the Geriatric Study Group of the Portuguese Internal Medicine Society. These groups need to cooperate in order to provide training and funding for scientific initiatives, as these objectives can only be accomplished with collaborative work.

Six Portuguese oncologists have taken part in the SIOG Advanced Course in Geriatric Oncology since this survey was performed (in addition to three more who had completed this course previously), and others were/are being trained in centers with geriatric oncology expertise in Europe, in addition to the educational sessions promoted by GTOG in the last two years.

The incorporation of geriatric evaluation into routine clinical practice is hampered by a number of factors, including a lack of time, a shortage of geriatricians and the absence of a national plan. However, we must not forget that a GA can change oncologic treatment plans, lead to non-oncologic interventions, increase the likelihood that a patient will complete their treatment, minimize the risk of complications and toxicity and improve their physical health and quality of life [26]. Accordingly, GA must serve as the focal point of any intervention and be used as part of the standard of care for older patients with cancer since it is just as crucial as recommending an oncologic therapy. This study also serves as a reminder that there is still more work to be done in order to put active aging-related policies and initiatives into practice.

## 5. Limitations

Limitations to this study that may be addressed in future research include the relatively small sample size, which does not allow a generalization of the conclusions. Although comparable to other studies of this kind, the low response rate (31.7%) makes it difficult to rule out a potential response bias in favor of individuals with an interest in geriatric oncology. The variations and similarities between the Portuguese setting and other European nations should be compared, especially with regard to how each nation develops its aging policies while taking into account international guidelines. Differences in the type of clinical practice (academic/university hospital, comprehensive cancer center or general hospital) were not evaluated in this questionnaire. Another limitation is the need to combine the perceptions and needs in geriatric oncology identified by healthcare professionals in the survey with a mapping of cancer and aging policies in Portugal. The Portuguese Oncology Society’s Geriatric Oncology Working Group will play a significant role in this area.

## Figures and Tables

**Figure 1 geriatrics-07-00091-f001:**
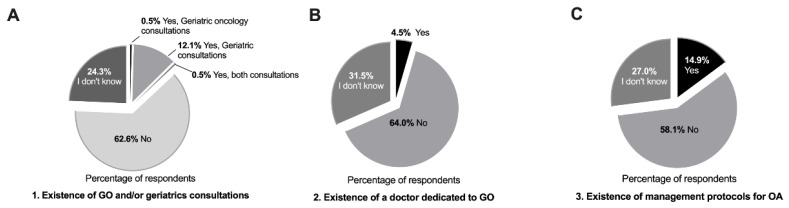
Geriatric oncology situation in Portuguese hospitals. (**A**) Frequency of geriatric oncology and/or geriatrics care in Portuguese hospitals. (**B**) Medical oncology departments with physicians practicing geriatric oncology. (**C**) Existence of specific management protocols for the care of elderly cancer patients. (**A**–**C**) Values are expressed as percentages, total number of answers (n = 222). Abbreviations: GO—geriatric oncology; OA—older adults.

**Figure 2 geriatrics-07-00091-f002:**
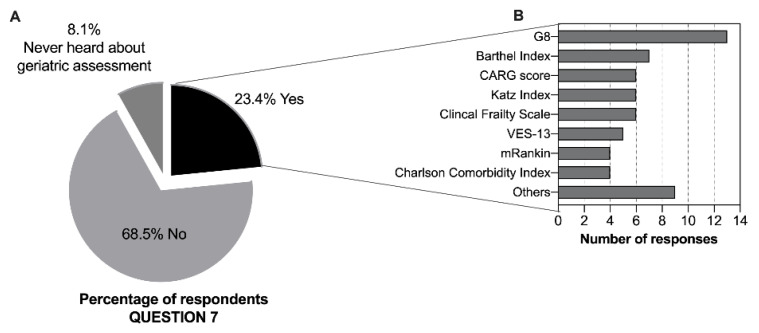
Geriatric assessment awareness and use of GA tools. (**A**) Reported use, in clinical practice, of GA/geriatric screening to evaluate elderly cancer patients (n = 222, values are expressed as percentages). (**B**) Geriatric assessment or screening tools used (n = 47 respondents, absolute values are reported). Abbreviations: G8 (Geriatric 8); CARG (Cancer and Aging Research Group); VES-13 (Vulnerable Elderly Survey-13); mRankin (modified Rankin) scale; others include: Timed Up and Go Test (TUG), Mini Mental State Examination, ePrognosis, Geriatric Depression Scale (GDS), Palliative Prognostic Index (PPI), Lawton–Brody Scale.

**Table 1 geriatrics-07-00091-t001:** Survey questions.

1. Does the hospital where you work offer any geriatric oncology and/or geriatrics consultations?
2. In the medical oncology service of the hospital where you work, is there a doctor specifically dedicated to geriatric oncology?
3. Does the hospital where you work have specific management protocols for elderly cancer patients?
4. From your clinical practice, do you perceive that the number of elderly cancer patients (>70 years) has increased?
5. In your opinion, do elderly cancer patients need more specific care when compared to younger patients?
6. Do you feel the need for assessment scales for elderly cancer patients, in addition to ECOG-Performance status and Karnofsky, to help you make treatment decisions?
7. In your clinical practice, do you use any geriatric assessment/screening to evaluate the elderly cancer patients (even if they are not validated for Portuguese language)?
8. Do you think that more information and training in geriatric oncology is needed?
9. How do you think geriatric assessment could help you in your clinical practice? You can choose more than one option.
10. What do you think is important to develop in the field of geriatric oncology in Portugal? You can choose more than one option.

**Table 2 geriatrics-07-00091-t002:** Respondent characteristics.

Characteristic	Total
**Number of participants, n**	222
**Age, median (years), (min–max)**	36 (78–24)
**Gender, n (%)**	
**Female**	151 (68.0)
**Male**	71 (32.0)
**Location, n (%)**	
**North**	140 (63.1)
**Center**	28 (12.6)
**South**	50 (22.5)
**Islands (Azores and Madeira)**	4 (1.8)
**Specialties**	
**Medical oncologist**	119 (53.6)
**Surgical specialty**	31 (14.0)
**Internal medicine**	27 (12.2)
**Radiation oncologists**	12 (5.4)
**Other specialties**	33 (14.9)

**Table 3 geriatrics-07-00091-t003:** Characteristics associated with the reported use of geriatric assessment.

Characteristic	Use of GA Tools	*p* Value
**Location, n (%)**		
** South**	18/50 (36.0)	0.054
** Other locations**	34/172 (19.8)	
**Specialty**		
** Medical specialties ***	18/58 (31.0)	0.009
** Medical oncology/Onco-hematology**	18/58 (31.0)	
** Surgical specialties ^#^**	2/32 (6.3)	
** Radiation oncologists**	0/12 (0.0)	

Total number of answers (n = 222). Abbreviations: GA—geriatric assessment. * Medical specialties included internal medicine, pneumology, gastroenterology. ^#^ Specialties included were general surgery, urology and otorhinolaryngology.

**Table 4 geriatrics-07-00091-t004:** Advantages of geriatric assessment and potential areas for the development of geriatric oncology.

Responses to Questions 9 and 10 ^#^	n (%)
**Q9. How do you think geriatric assessment could help you in your clinical practice?**	
**To define a treatment strategy**	189 (85.1)
**To detect frailty**	172 (77.5)
**To predict toxicity**	163 (73.4)
**To improve quality of life**	163 (73.4)
**To predict survival**	83 (37.4)
**I do not think GA would help in my clinical practice**	1 (0.5)
**Q10. What do you think is important to develop in the field of geriatric oncology in Portugal?**	
**Systematic GA in oncology services**	178 (80.2)
**Invest in training in geriatrics both at the undergraduate and postgraduate level**	156 (70.3)
**Geriatricians to be part of multidisciplinary teams**	106 (47.7)
**Creation of study groups in geriatric oncology**	102 (45.9)
**Creation of geriatric oncology units**	68 (30.6)
**I do not believe anything is necessary**	1 (0.5)

^#^ For these questions, more than one option could be chosen. Total number of answers (n = 222).

## Data Availability

The data presented in this study are available on request from the corresponding author.

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
