# Peer review of "Geriatric Oncology in Portugal: Where We Are and What Comes Next—A Survey of Healthcare Professionals"

_geriatrics, 2022, doi:10.3390/geriatrics7050091_

Round 1

Reviewer 1 Report

This study uses a survey methodology to conduct a national survey. This addresses practice, needs, and gaps in the growing area of Geriatric Oncology. Although, the types of questions are not novel in themselves, they are novel in the Portuguese setting. The method is appropriate to the research question. The response rate is higher than in many other studies of this type. There are a few minor spelling and grammar errors, however, overall the paper is clearly written.

Introduction

A geriatric audience may not be familiar with Geriatric Oncology. A brief paragraph explaining how GA assessment and intervention might be delivered in Cancer Care/some Geriatric Oncology models of care would be informative for such readers.

The definition of GA on p 2 (line 44-45) defines it as having a “focus on medical, physiological and functional capability” in older patients. GA also typically includes psycho-social domains.

There are a couple of very long sentences (line 36-41 on p1) which might be easier to read if separated into a greater number of sentences.

Methods

As the survey instrument is included in a table, the description of the survey instrument p3, lines 75-86 could be more concise.

Page 2, line 67 should read other instead of “rather”.

In page 2, line 70, I am not sure what the “and was accessible for answer” phase is referring to, it may be superfluous.

The data analysis is well described.

Results

Some results are duplicated in text and tables, and thus the text could be more concise, focusing on summarising the findings and additional information not in the tables and figures.

A geriatric audience may not be familiar with the tools VES-13, G8 and CARG. These (and the other tools mentioned) should be referenced.  

Figure 1 would be easier to understand quickly if each Pie Chart was labelled with the topic rather than the question number.

Discussion

It would be interesting to see access to geriatricians/Geriatric oncology in this study compared with reports from surveys in other countries. Also a comparison of Geriatrician numbers would also be of interest.

Author Response

Reviewer 1

This study uses a survey methodology to conduct a national survey. This addresses practice, needs, and gaps in the growing area of Geriatric Oncology. Although, the types of questions are not novel in themselves, they are novel in the Portuguese setting. The method is appropriate to the research question. The response rate is higher than in many other studies of this type. There are a few minor spelling and grammar errors, however, overall the paper is clearly written.

Introduction

Point 1. A geriatric audience may not be familiar with Geriatric Oncology. A brief paragraph explaining how GA assessment and intervention might be delivered in Cancer Care/some Geriatric Oncology models of care would be informative for such readers.

Response to point 1: As suggested a brief paragraph explaining geriatric models of care was added to the manuscript in page 2 lines 54-58:

“A multidisciplinary consultative GA, a geriatrician embedded within an oncology clinic, and primary management by a dual-trained geriatric oncologist are just a few examples of the various models for integrating geriatrics into oncology care that exist (reviewed in [5]). The ideal model does not exist, and GA should be flexible and tailored to the resources available”

Point 2: The definition of GA on p 2 (line 44-45) defines it as having a “focus on medical, physiological and functional capability” in older patients. GA also typically includes psycho-social domains.

Response to point 2: The authors agree with the reviewer, as the definition of GA was not complete, and we have now rephrase it as follows (page 2 , lines 49-51):

“GA is a multidimensional interdisciplinary diagnostic process with a focus on medical, physiological, functional and psycho-social domains, in older vulnerable or frail patients, in order to identify impairments that are not routinely detected during usual oncology consultations . It also includes a coordinated and integrated plan for treatment and follow-up”.

Point 3: There are a couple of very long sentences (line 36-41 on p1) which might be easier to read if separated into a greater number of sentences.

Response to point 3: The authors have now re-written this section for an easier reading. Please see (line 36-41 on p1).

Methods

Point 4: As the survey instrument is included in a table, the description of the survey instrument p3, lines 75-86 could be more concise.

Response to point 4: As suggested this section was simplified. Please see subsection of Methods (2.1. Survey development and setting)

Point 5: Page 2, line 67 should read other instead of “rather”.

Response to point 5: The authors have now corrected this spelling error.

Point 6: In page 2, line 70, I am not sure what the “and was accessible for answer” phase is referring to, it may be superfluous.

Response to point 6: The authors have now removed this sentence from the manuscript.

The data analysis is well described.

Results

Point 7: Some results are duplicated in text and tables, and thus the text could be more concise, focusing on summarising the findings and additional information not in the tables and figures.

Response to point 7:  As suggested the authors have now removed some data clearly presented in tables and figures. Please refer to tracking changes in the subsection 3.2. Geriatric Assessment and Screening for a more detailed analysis.

Point 8: A geriatric audience may not be familiar with the tools VES-13, G8 and CARG. These (and the other tools mentioned) should be referenced.  

Response to point 8: As suggested the authors have now added the references (see page 4 lines 148-149).

Point 9: Figure 1 would be easier to understand quickly if each Pie Chart was labelled with the topic rather than the question number.

Response to point 9: As suggested the authors have now used topic as label, instead of question number to facilitate reading and interpretation (please see page 4 figure 1).

Discussion

Point 10: It would be interesting to see access to geriatricians/Geriatric oncology in this study compared with reports from surveys in other countries. Also a comparison of Geriatrician numbers would also be of interest.

Response to point 10: As suggested by the reviewer we have now added in the discussion comparison to other studies, such as a Mexican nationwide survey and a US study. We have also compared our results with the Spanish study from SEOM and a survey performed in the Netherlands.

The reviewer can now read in the discussion:

Page 6 line 205-208: “ In Spain, according to Spanish Society of Geriatrics and Gerontology, there are approximately 2456 physicians involved in geriatric care (970 are geriatricians), and 33 accredited centers to train geriatricians [23, 24].”

Page 6 lines 222-229: “ The proportion of physicians who reported performing a GA is comparable to that found in other studies, such as a survey of cancer providers in the United States (US) (21%)[4] or a nationwide survey in Mexico (18.9%)[20]. Studies from European countries, as Spain, revealed a 31% use of GA in clinical practice [21], and an older study from the Nether-lands, showed that 60% of healthcare professionals performed some sort of geriatric evaluation [27]. Because Europe has the largest population share of older adults world-wide, it is not surprising the increasing awareness of the need for geriatric oncology practices in this continent.”

Page 6 lines 235 – 239: “Global geriatric oncology initiatives are revolutionizing the way elderly cancer patients are being treated [2, 28]. Particularly in Europe, where there have been a rising number of new specialized clinics, initiatives to enroll in clinical trials and joint initiatives to develop clinical trials for older adults with cancer[3].”

Reviewer 2 Report

The research is a cross sectional study about geriatric oncology in Portugal .

The method is well described, as it is it seems that the questionnaire was send only once ? That may be explain the low answer rate (222/700).This could be a biais for the study as probably only those  interested to the geriatric oncology responded .This should be added to the limitation section .

in the discussion part , the autors compare the results  the others countries, but I think that you should insert the numbers  fund in the results of the studies cited . It seems that there is only non- european study, did you check with the european society ?

Author Response

Reviewer 2

The research is a cross sectional study about geriatric oncology in Portugal .

Point 1: The method is well described, as it is it seems that the questionnaire was send only once ? That may be explain the low answer rate (222/700).This could be a biais for the study as probably only those  interested to the geriatric oncology responded .This should be added to the limitation section .

Response to point 1: The questionnaire was send twice by email, separated by 1 month.

Although the absolute number of responses seems low, the response rate (31.7%) is higher than in many other studies of this type. The survey from the Spanish Geriatric Oncology Group was answered by 154 medical oncologists (ref1). Although they do not refer the total target population, the number of physicians in Spain is considerably higher than in Portugal. Additionally, a geriatric oncology survey from Australia reported a response rate of 11% (n=69 oncologists completed the survey) (ref2). A survey performed in the Netherlands reported an overall response rate of 34% (n = 183) (ref3).

Additionally a ESMO-SIOG survey about geriatric oncology pratices reported last year at ESMO congress, was answered by a total of 168 participants  (ref 4).

Therefore we believe that response rate is not a limitation of our study.

For more details please check the references:

  1. Gironés R, Morilla I, Guillen-Ponce C, Torregrosa MD, Paredero I, Bustamante E, Del Barco S, Soler G, Losada B, Visa L, Llabrés E, Fox B, Firvida JL, Blanco R, Antonio M, Aparisi F, Pi-Figueras M, Gonzalez-Flores E, Molina-Garrido MJ, Saldaña J; Spanish Working Group on Geriatric Oncology of the Spanish Society of Medical Oncology (SEOM). Geriatric oncology in Spain: survey results and analysis of the current situation. Clin Transl Oncol. 2018 Aug;20(8):1087-1092.)
  2. To THM, Soo WK, Lane H, Khattak A, Steer C, Devitt B, Dhillon HM, Booms A, Phillips J. Utilisation of geriatric assessment in oncology - a survey of Australian medical oncologists. J Geriatr Oncol. 2019 Mar;10(2):216-221.
  3. Jonker JM, Smorenburg CH, Schiphorst AH, van Rixtel B, Portielje JE, Hamaker ME. Geriatric oncology in the Netherlands: a survey of medical oncology specialists and oncology nursing specialists. Eur J Cancer Care (Engl). 2014 Nov;23(6):803-10.
  4. Baldini, C. et al.1827P European Society for Medical Oncology (ESMO)/International Society of Geriatric Oncology (SIOG) Joint Working Group (WG) survey on management of older patients with cancer. Annals of Oncology, Volume 32, S1237 - S123

Point 2: in the discussion part , the autors compare the results  the others countries, but I think that you should insert the numbers  fund in the results of the studies cited . It seems that there is only non- european study, did you check with the european society ?

Response to point 2: As suggested by the reviewer we have now added in the discussion comparison to other studies, such as a Mexican nationwide survey and a US study. We have also compared our results with the Spanish study and a survey performed in the Netherlands, and other European data.

The reviewer can now read in the discussion:

Page 6 line 205-208: “ In Spain, according to Spanish Society of Geriatrics and Gerontology, there are approximately 2456 physicians involved in geriatric care (970 are geriatricians), and 33 accredited centers to train geriatricians [23, 24].”

Page 6 lines 222-229: “ The proportion of physicians who reported performing a GA is comparable to that found in other studies, such as a survey of cancer providers in the United States (US) (21%)[4] or a nationwide survey in Mexico (18.9%)[20]. Studies from European countries, as Spain, revealed a 31% use of GA in clinical practice [21], and an older study from the Nether-lands, showed that 60% of healthcare professionals performed some sort of geriatric evaluation [27]. Because Europe has the largest population share of older adults world-wide, it is not surprising the increasing awareness of the need for geriatric oncology practices in this continent.”

Page 6 lines 235 – 239: “Global geriatric oncology initiatives are revolutionizing the way elderly cancer patients are being treated [2, 28]. Particularly in Europe, where there have been a rising number of new specialized clinics, initiatives to enroll in clinical trials and joint initiatives to develop clinical trials for older adults with cancer[3].”

Page 7 lines 249-252: "Most of the needs in geriatric oncology identified by Portuguese respondents are common among European countries, as recently revelead by an ESMO-SIOG Joint Working Group short survey on management of older patients with cancer [30]."